# Flavin-Conjugated Iron Oxide Nanoparticles as Enzyme-Inspired Photocatalysts for Azo Dye Degradation

**Samer I. Nehme** **, Leander Crocker and Ljiljana Fruk \***

Department of Chemical Engineering and Biotechnology, University of Cambridge, Phillipa Fawcett Drive, Cambridge CB3 0AS, UK; sin20@cam.ac.uk (S.I.N.); lbc32@hermes.cam.ac.uk (L.C.)

**\*** Correspondence: lf389@cam.ac.uk

**Abstract:** In this work, a new photocatalytic system consisting of iron oxide nanoparticles (IONPs), coated with a catechol-flavin conjugate (DAFL), is synthesized and explored for use in water remediation. In order to test the efficiency of the catalyst, the photodegradation of amaranth (AMT), an azo dye water pollutant, was performed under aerobic and anaerobic conditions, using either ethylenediaminetetraacetic acid (EDTA) or 2-(*N*-morpholino)ethanesulfonic acid (MES) as electron donors. Depending on the conditions, either dye photoreduction or photooxidation were observed, indicating that flavin-coated iron-oxide nanoparticles can be used as a versatile enzyme-inspired photocatalysts.

**Keywords:** water remediation; photoreduction; photooxidation; azo dye; flavin; iron oxide nanoparticles; charge transfer; reactive oxygen species; electron donor

## 1. Introduction

The treatment of industrial wastewater contaminated with synthetic organic compounds from the textile, cosmetic, pharmaceutical and other industries, is one of the most important tools in our fight against water pollution [1–4]. The textile industry specifically is one of the largest contributors to aquatic pollution. In fact, 17%–20% of the industrial contamination of water can be accredited to dyes being used in the textile industry [5]. Moreover, 10%–15% of these dyes are released into the environment without any pre-treatment [6] and many of them have been shown to be toxic to aquatic life [5].

The majority of the synthetic aromatic dyes used in the textile industry are azo dyes. They make up around two-thirds of all synthetic dyes used [7], and the demand for azo dyes such as amaranth (AMT) continues to increase [8]. Azo dyes are complex aromatic molecules characterized by one or multiple azo bonds (–N=N–), which are the main contributors to the toxicity of these dyes, in particular their cancerogenic properties [9]. For instance, the azo dye 4-aminoazobenzene (aniline yellow) was shown to display liver toxicity in male mice, in doses as small as 0.027–0.15 µmol/g body weight [10]. This dye has also been shown to induce tumor formation in rats when given orally or applied onto the skin [7]. Furthermore, the complexity of many azo dyes' structure results in nonbiodegradability and non-biocompatibility.

Due to their demonstrated long term toxicity, many strategies have been developed and explored to degrade azo dye pollutants into less toxic products, among them some conventional chemical and physical techniques, such as coagulation, filtration, flocculation, adsorption and ozonation [11,12]. However, these strategies, besides being expensive and slow, often do not lead to the complete removal or degradation of the dyes.

Studies have also shown that consuming water which has been treated with some of the conventional purification strategies can lead to carcinogenesis in humans [5].

A promising milder water treatment strategy, in particular for the removal of polluting dyes, is the use of micro-organisms. A wide variety of micro-organisms, such as bacteria (*C. perfringens*) [13], a white-rot fungus [14] and yeast [9] produce enzymes capable of azo bond degradation. These microorganisms and their isolated enzymes allow for degradation in both the presence and absence of oxygen [15,16]. A key enzyme class involved in azo dye degradation is the family of azoreductases found within *E. coli*, which require both the flavin mononucleotide (FMN) and NAD(P)H cofactors for the reaction completion [17]. Azoreductases have been identified to reduce azo bonds via the consumption of two reducing equivalents of NADH, resulting in colorless aromatic amines, as shown in (Scheme 1). [5,17,18]

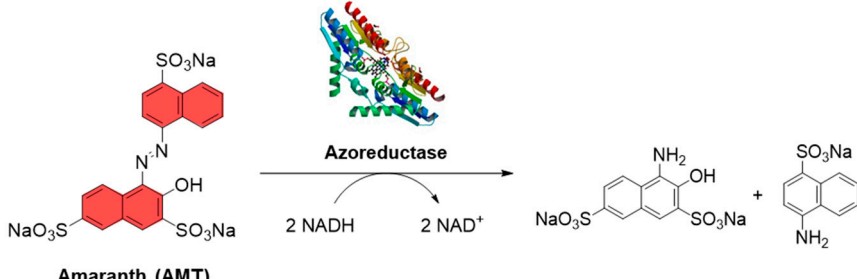

**Scheme 1.** The reduction of the amaranth (AMT) azo bond by an azoreductase enzyme (AzoR).

Although enzymes are specific and environmentally friendly catalysts, their application in water remediation is hindered due to their sensitivity to temperature and instability in alkaline conditions [5]. One strategy to overcome these limitations is to use a catalytic component, an organic cofactor instead of the whole enzyme, and repurpose it to afford the dye degradation, but removing the need for tightly controlled conditions and the use of additional cofactors such as nicotinamide dinucleotide (NADH). We initially explored such an approach by taking an advantage of the photocatalytic activity of flavin [19], and combining it with a polydopamine carrier, which acts as an active cofactor support. Such stabilized flavin was successfully used to mimic enzyme-like reactions using light instead of NADH [20,21]. Herewith, we report on a design of a catalyst that combines flavin and semiconducting, magnetic nanoparticles to afford a multifunctional system capable of effective dye degradation, while being easy to prepare and separate from the reaction mixture.

Semiconductor-based photocatalysts, such as titanium dioxide ($TiO_2$), zinc oxide (ZnO) [22,23], tungstate ($WO_3$) [24], vandate ($VO_4$) [25], and others [26,27] have already been successfully used for the removal of organic pollutants from water. Although they have high efficiency, as well as photostability, and some, such as $TiO_2$ nanoparticles, are cheap to produce [8,12,28], they are difficult to separate and reuse. For example, wastewater remediation in bulk, using photocatalytically-active $TiO_2$ nanoparticles, still requires the filtration of the catalyst after the photodegradation is complete. One way around this problem is immobilizing the catalyst onto a solid support that can be removed from the solution [12]. However, this can also be expensive and inconvenient on large scales.

A viable alternative would be using magnetic nanoparticles, such as iron oxides (IONPs) that can be retrieved from solution with the help of magnets. Besides being magnetic, IONPs have some other advantages over $TiO_2$ and other conventionally used photocatalytic metal-oxide nanoparticles. For example, IONPs have a band gap of 2.2–2.3 eV, which is lower than most commonly used semiconducting particles [29]. As a result, the particles can absorb light from the visible spectrum range, and do not have to rely on higher energy UV light for photocatalysis. Sundaramurthy et al. has shown that IONPs can photocatalytically degrade Congo red (CR) dye through formation of reactive oxygen species (ROS), with the help of photoexcited electrons [30]. In addition, they are easy to prepare on a larger scale, and are shown to be biocompatible, and have extensively been used in medicine [31].

However, the main disadvantage of IONPs photocatalysts is their high degree of photoexcited electron-hole recombination [32], but this could be overcome by anchoring an appropriate photosensitizer to the particle surface.

Herein, we present a novel heterogeneous photocatalytic system comprised of $\gamma$-Fe$_2$O$_3$ IONPs functionalized with a novel catechol-flavin compound (DAFL). Such a IONP-DAFL hybrid system can afford efficient azo bond degradation in both aerobic and anaerobic conditions (Scheme 2), but this is a feature not widely exemplified in the literature for other photocatalytic systems. The catechol moiety coupled onto flavin species enables both an efficient flavin anchoring to the IONP surface and improved electron transfer between the flavin and $\gamma$-Fe$_2$O$_3$. The hybrid catalytic system is efficient, easy to prepare, and separate from the solution, and can be successfully used in the removal of water-polluting dyes.

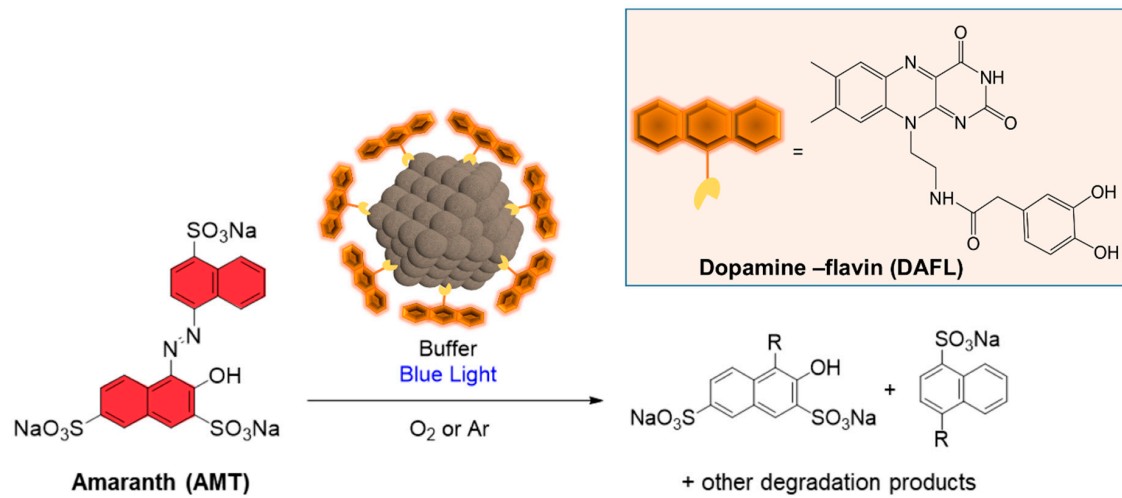

**Scheme 2.** The degradation of amaranth (AMT) by hybrid iron oxide-flavin (IONP-DAFL) photocatalyst under irradiation with blue light. Structure of dopamine-flavin derivative is shown in inset.

Some examples of the flavin-conjugated metal-oxide surface such as TiO$_2$ and BiOCl have been reported, but not applied to the degradation of complex molecules, such as AMT, and to the best of our knowledge, none demonstrated both the reduction and the oxidation of the substrates as presented within this paper [33–35].

## 2. Results and Discussion

### 2.1. Synthesis of $\gamma$-Fe$_2$O$_3$ Maghemite Iron Oxide Nanoparticles (IONPs)

The $\gamma$-Fe$_2$O$_3$ IONPs were prepared using an oil in water (*o/w*) reverse micro-emulsion protocol, as reported by Benyettou et al. [36] using ferrous dodecyl sulfate (Fe(DS)$_2$). Prepared IONPs were first characterized using Attenuated total reflection Fourier transform IR (ATR-FTIR) and TEM (Figure 1).

Figure 1A shows bands at 962 cm$^{-1}$ and 1619 cm$^{-1}$ in the FTIR fingerprint region of the Fe(DS)$_2$ spectrum that correspond to the S=O stretching vibrations of the sulfonic acid [37]. The sharp peaks at ~2919 cm$^{-1}$ relate to the C–H stretching mode. The band at 3415 is a characteristic of the O–H bond vibration from the residual water that remained in the dried sample. The FTIR spectrum of the IONP shows characteristic bands at 556 cm$^{-1}$ and 3270 cm$^{-1}$ that, respectively, correspond to the Fe–O stretching vibration and the O–H stretching bond vibrations from the residual water molecules coordinated to the particle surfaces [38]. The absence of the strong, sharp peaks in the fingerprint region of the IONP spectrum indicates that only trace amounts of surfactant molecules are present [39].

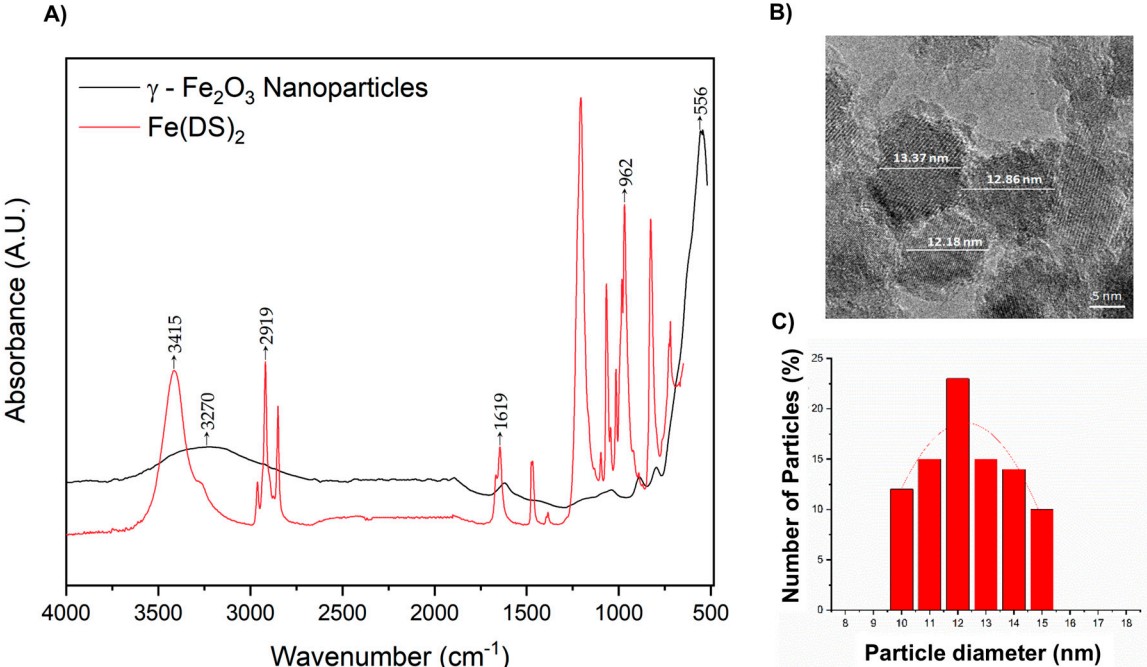

**Figure 1.** (**A**) The Attenuated Total Reflection Fourier-Transform Infra-Red (ATR-FTIR) spectroscopy spectrum of the $\gamma$-Fe$_2$O$_3$ iron oxide nanoparticles (IONPs), in comparison to the Fe(DS)$_2$ starting material. (**B**) Transmission electron microscopy (TEM) images of IONPs. (**C**) The particle size distribution determined from the average of ~100 measurements of particles.

The zeta potential and hydrodynamic size of the nanoparticles were investigated using a Zetasizer Nano Range instrument (see Table S1). At a pH ~7, the IONPs have a surface charge of −19.25 ± 4.51 mv. The negative charges, due to hydroxyl groups on the nanoparticle surfaces, stabilize the suspension and stop the particles from aggregating. The IONPs displayed an average hydrodynamic size of 152.3 ± 2.45 nm, with a polydispersity index (PDI) of 0.211 ± 0.014 from dynamic light scattering (DLS) measurements. Transmission electron microscopy (TEM) was also used to study the IONPs size, distribution and morphology (Figure 1B,C). The data shows rough spherical particles with an average diameter of 12.38 ± 1.12 nm. Particle diameters range from 10 to 15 nm with 12 nm, particles being the most frequently occurring. This disparity in particle size from DLS and TEM techniques is likely due to hydrate layers present on the IONPs in an aqueous medium [40] and the nanoparticle agglomeration in water due to magnetic attraction [41].

X-ray diffraction (XRD) 2θ values for the IONPs align with literature standards [42] (Figure S1). Diffraction peaks at (220), (311), (222), (400), (422), (511) and (440) are indicative of a cubic close-packed (CCP) spinel crystal lattice. However, since both magnetite and maghemite have similar lattice structures, distinguishing the two phases from XRD alone is difficult [43]. The valency of the Fe atoms determined through XPS can help determine the different crystalline phases of the metal oxide.

## 2.2. Surface Modification of $\gamma$-Fe$_2$O$_3$ Nanoparticles

Although the functionalization of IONPs is well documented in the literature [44,45], the use of a catechol linker such as dopamine for functionalization, while avoiding the generation of the polydopamine coating, which could interfere with the electron transfer, has been reported less frequently. Herewith, a modified protocol described by Geiseler et al. [46] was used for the functionalization of the IONP surfaces with dopamine (DA) or dopamine-flavin (DAFL, see electronic supplementary information (ESI) for details on the synthesis). The dopamine-based linker was added to an IONP suspension at 23 °C (0.1 mg of DAFL for every 1 mg of IONP).

Since DA polymerizes into polydopamine (PDA) in the presence of $O_2$, light and at basic pH [47], the linker coordination process was conducted in an Ar atmosphere, in the dark, and at a pH of ~7. The coated IONPs were kept in the dark at room temperature in air.

ATR-FTIR and TEM were used to characterize IONP-DAFL (Figure 2) and the IONP-DA control (Figures S2 and S3). The ATR-FTIR spectrum of DAFL (Figure 2A) shows a sharp peak at 1535 cm$^{-1}$ that is characteristic of flavin C=N modes of the isoalloxazine ring [48]. Moreover, combined bond vibrations of C=N and C=C are present at 1258 cm$^{-1}$ [20]. These bond vibrations are also present in the IONP-DAFL FTIR spectrum, indicating a successful coordination of the linker on the particle surfaces.

Furthermore, the band at 1388 cm$^{-1}$ corresponds to the stretching vibration of the C–N bond, connecting the catechol to the flavin moiety. The vibrations at 647 cm$^{-1}$ and 928 cm$^{-1}$ correspond to the aromatic C–H bond vibrations [49]. Lastly, the peaks at 3241 cm$^{-1}$ and 559 cm$^{-1}$ in the IONP-DAFL spectrum correspond to the O–H and Fe–O bond vibrations, respectively.

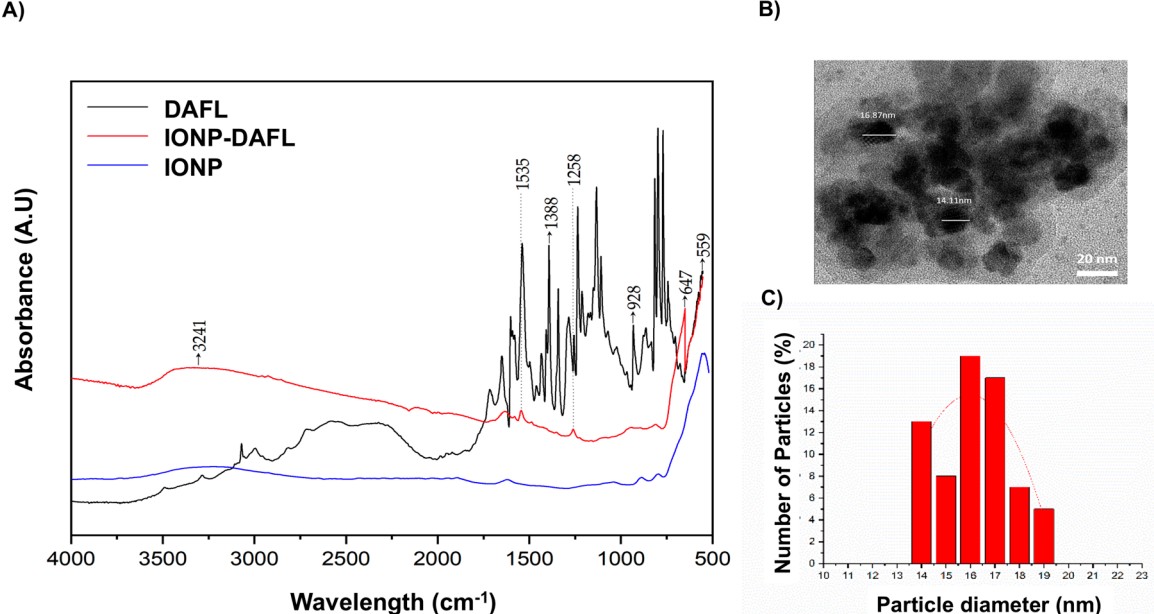

**Figure 2.** (**A**) The ATR-FTIR spectra of IONP-DAFL, IONP, and dopamine-flavin (DAFL). (**B**) TEM images of IONP-DAFL. (**C**) The particle size distribution determined from the average of ~100 measurements of particles.

After functionalization with DA and DAFL, the zeta potential of the IONPs increases from −19.25 ± 4.51 mv to 27.33 ± 0.22 mV and 17.93 ± 1.04 Mv, respectively (Table S1). The shift from a negative to positive charge for both modified IONPs indicates successful surface functionalization, due to the introduction of amino moieties within the linker molecules. The size of the IONPs also increases from 152.3 ± 2.5 nm (PDI = 0.211) to 196.4 ± 4.7 nm (PDI = 0.320) for IONP-DA and 209.5 ± 1.4 nm (PDI = 0.320) for IONP-DAFL. This increase of hydrodynamic radius provides further evidence of successful catechol coating.

Sizes obtained from DLS are compared to TEM images of the IONP-DAFL (Figure 2B) and IONP-DA (Figure S3). IONP-DAFL particles are spherical in shape, and have an average diameter of 16.17 ± 0.86 nm. They range in size from 14 to 19 nm, with 16 nm particles being the most frequently occurring. IONP-DA nanoparticles, on the other hand, have an average size of 17.69 ± 0.94 nm, with particles ranging from 15 to 20 nm in diameter. Thus, the size of both surface-modified IONPs is roughly the same.

In order to confirm the presence and determine the loading percentage of DAFL on the IONP surface, XPS analysis was utilized (Figure S4). The XPS survey spectrum (Figure S4A) shows the presence of Fe, O, N and C, as expected. The atomic % of N was used to calculate the % loading of

DAFL, as there are 5 N atoms per DAFL molecule, we can therefore assume the loading to be 1%. The high-resolution Fe 2p spectrum (Figure S4B) shows two distinct peaks with binding energies of 710.9 eV for Fe $2p_{3/2}$ and 724.5 eV for Fe $2p_{1/2}$ [50]. The satellite peaks present at 718.8 eV and 732.7 eV are characteristic for $Fe^{3+}$ ions in $Fe_2O_3$ [51]. The fitting also gives more detail as to the IONP composition, which shows both $Fe^{3+}$ and $Fe^{2+}$ ions present with the larger amount of $Fe^{3+}$. This has been observed for other $Fe_2O_3$ nanostructures containing both α- and γ-phase $Fe_2O_3$ [52]. The O 1s spectrum clearly shows the characteristic signals one would expect for the IONP-DAFL hybrid, including lattice $Fe_2O_3$ at 530.1 eV, carbonyl C=O at 531.1 eV, a lattice hydroxyl signal at 532.7 eV that is commonly observed in $Fe_2O_3$, and another peak at around 536.6 eV, which could be ascribed to the catechol Fe–O bond [53]. The nitrogen 1s spectrum clearly shows a main C–N/C=N bond signal at 400.0 eV. Finally, the C 1s spectrum displays all characteristic signals associated that correspond to DAFL, including C–C at 285.0 eV, C–O bonding at 286.3 eV, carbonyl C=O at 288.3 eV, and the π–π* satellite can be observed at 290.4 eV.

Finally, as expected, the XRD spectrum of IONP-DAFL shows peaks at the same 2θ values as IONP. Overlapping diffraction peaks at (220), (311), (222), (400), (422), (511) and (440) indicate that the CCP spinel lattice structure was not changed through the process of surface-coating the particles (Figure S5).

### 2.3. Amaranth Photodegradation

#### 2.3.1. Amaranth Photooxidation (Aerobic Activity)

The photocatalytic activity of IONP-DAFL towards AMT degradation was first carried out in an aerobic environment. In this condition, dye degradation is dependent upon the formation of reactive oxygen species (ROS) during the irradiation process [8]. These ROS are produced by semiconductors, such as IONP, through electron-hole (e-h) pair generation upon irradiation with photons with energy equal to or greater than the band gap energy of the material [54]. If the charge separation is maintained and the electron and holes do not recombine, electrons migrate to the conduction band, leaving a vacancy (a hole) in the valence band. The electron and hole can then recombine with $O_2$ and $H_2O$, resulting in the production of the superoxide and hydroxyl radicals, as shown in Scheme 3. This process is limited by the rate of electron-hole recombination, which has been shown to be high for IONPs [29], so that a suitable sacrificial electron donor (ED) needs to be employed to quench generated holes, and an additional photoactive 'sensitizer' is required to improve activity. We envisioned that the addition of flavin-catechol, DAFL, would act as an IONP stabilizer and improve the visible-light IONPs catalysis. This is due to the direct charge transfer of electrons from flavin to the particle, as well as the enhanced ROS production by intermolecular charge and energy transfer to molecular oxygen, which results in the production of superoxide and singlet oxygen (Scheme 3). The direct charge transfer from flavin to NP through the catechol anchoring group could be expected as we observed a large degree of fluorescence quenching in dopamine-bound flavin (DAFL) in comparison to the NBoc-protected flavin, **3** (Figure S6). The same phenomena was observed for other flavin-catechol conjugates that we have prepared [21].

In order to explore the efficiency of our system we chose two different classes of sacrificial electron donors: 2-(*N*-morpholino)ethanesulfonic acid (MES) at pH 6, a zwitterionic buffer recently shown to have favorable electron donor properties with flavins [55], and ethylenediaminetetraacetic acid (EDTA) at pH 6, which is one of the most widely used electron donors for flavin photoreduction [56]. The electron donor was used in excess (0.1 M) to AMT and IONP-DAFL, since the rate of the reaction has been shown to be proportional to the concentration of the donor present [57].

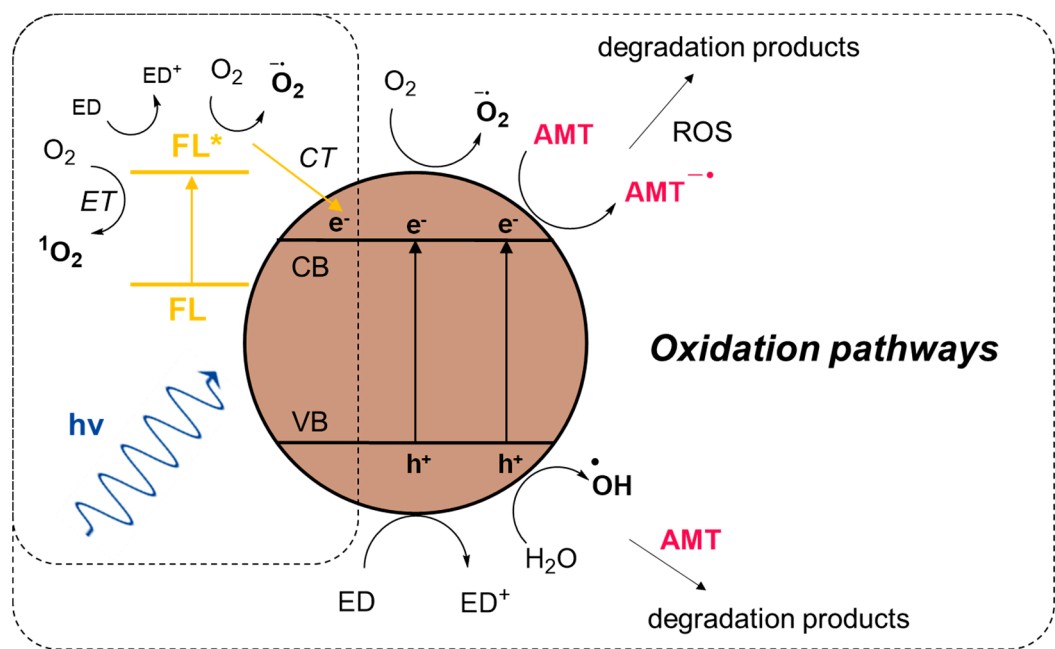

**Scheme 3.** Possible oxidation pathways of AMT degradation. Flavin photoexcitation results in charge transfer (CT) to the conduction band (CB) of the IONPs in the presence of an electron donor (ED), as well as energy transfer (ET) and charge transfer processes with molecular oxygen to produce reactive oxygen species (ROS), singlet oxygen and superoxide, respectively, which then go on to degrade AMT. Photoexcitation of IONP electron-hole (e-h) pairs can interact with $O_2$ to produce further superoxide radicals. AMT can also undergo a redox processes with these electron hole pairs initiating degradation through interactions with ROS.

As a control, dopamine (DA) was attached to the surface of IONP in order to observe any catechol-mediated activity [58]. After 3 h of irradiation with MES, the $C/C_0$ of AMT decreased by only 30% in the presence of the IONP (0.333 mg/mL) and IONP-DA (0.333 mg/mL) controls (Figure 3 A), showing that the catechol coating does not significantly enhance IONP photocatalytic activity, nor quenches ROS liberated from the IONP upon photoexcitation. However, both controls show much less activity in the presence EDTA after 3 h of irradiation with 450 nm blue light ($\lambda_{max}$ = 520 nm), proving that in the case of these controls, MES acts as a better electron donor than EDTA. (Figure 3B)

IONP-DAFL (0.333 mg/mL) and DAFL (0.003 mg/mL), on the other hand, degrade AMT by 91% and 60%, respectively, after 1 h of irradiation in the presence of MES. This clearly demonstrates the enhanced activity of the heterogeneous flavin-conjugated IONPs over the homogenous flavin alone. After 1 h of irradiation using EDTA, AMT is degraded by 86% in the presence of IONP-DAFL, slightly lower than that compared to MES, and surprisingly, by 98% in the presence of DAFL (see Figures S7 and S8 for UV-vis absorption spectra of AMT at each time point). This observation can be explained by the chemical nature of the electron donor used. It has been reported that photooxidation of EDTA leads to radical oxidation products that can degrade a substrate and the catalyst that generated them in the first place [59]. On the other hand, morpholine-based buffers, such as MES, not only act as electron donors, but have also been shown to inhibit oxidative flavin degradation through the formation of spin-correlated ion pairs that impede deleterious ROS [55]. In our case, this is clearly demonstrated by the relative activity of DAFL in MES and EDTA, the latter being much more efficient, most likely due to the additive effect of EDTA oxidation products degrading AMT. However, the combined IONP-DAFL system shows slightly better activity in the presence of MES, which can be attributed to the photoprotective effect of the morpholine buffer, which inhibits flavin-induced ROS formation, thus enhancing charge transfer to the IONP. For the rate and the quantum efficiency of the reaction, refer to the 'Quantum Efficiency' section of the ESI.

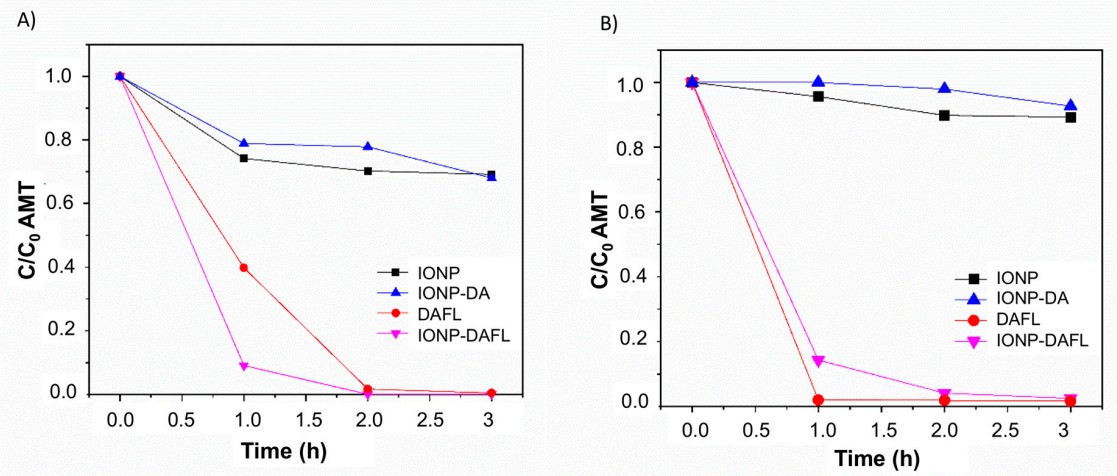

**Figure 3.** The relative concentration of the remaining AMT in solution ($C/C_0$) measured using UV-Vis absorption spectroscopy ($\lambda_{max}$ = 520 nm) after irradiation with 450 nm blue light for 3 h in the presence of IONP-DAFL, DAFL, IONP, or IONP-DA in (**A**) MES (0.1 M) and in (**B**) ethylenediaminetetraacetic acid (EDTA) (0.1 M) under an $O_2$ environment.

Finally, in order to study the nature of the ROS being produced, an antioxidant assay using ROS quenchers was designed. Mannitol, (2,2,6,6-tetramethylpiperidin-1-yl)oxidanyl (TEMPO), and 1,4-diazabicyclo [2.2.2]octane (DABCO) have been shown to scavenge hydroxyl radicals, superoxide and singlet oxygen species, respectively [54,60]. 1.5 molar equivalents of each scavenger are added independently to three different vials, each containing 0.5 mL of AMT (0.1 mg/mL), 1 mL of 0.333 mg/mL IONP or IONP-DAFL, and 1 mL of MES (0.1 M). Since MES is a more appropriate electron donor for IONP-DAFL in an aerobic environment, it was chosen for this experiment. The reaction mixture is flushed with $O_2$ before being irradiated for 3 h with aliquots being taken hourly, and analyzed (Figure 4). The UV-vis absorption spectra at each time point for IONP and IONP-DAFL can be found in Figures S9 and S10 of the ESI, respectively.

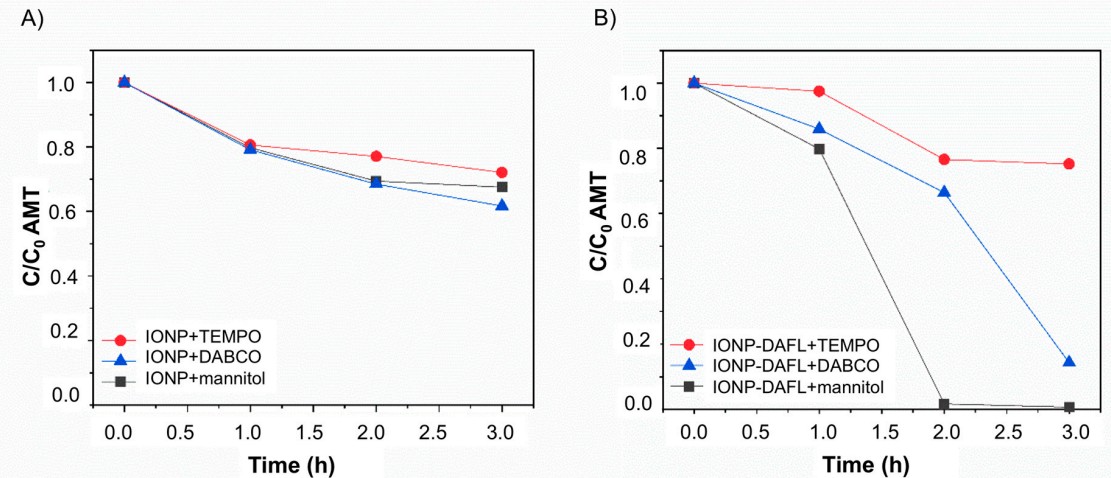

**Figure 4.** (**A**) The relative concentration of remaining AMT in solution ($C/C_0$) measured using UV-vis absorption spectroscopy ($\lambda_{max}$ = 520 nm) after irradiation with 450 nm blue light for 3 h in the presence of (**A**) IONP or (**B**) IONP-DAFL and MES under an $O_2$ atmosphere with (2,2,6,6-tetramethylpiperidin-1-yl)oxidanyl (TEMPO), 1,4-diazabicyclo [2.2.2]octane (DABCO) and mannitol as reactive oxygen species (ROS) scavengers.

The scavengers are expected to quench the ROS being produced in solution, thereby slowing the reaction down. Therefore, the ROS being produced can be identified by the effects observed when a scavenger is added to the reaction.

The addition of ROS scavengers to the bare IONP samples does not affect the particles' ability to degrade the AMT. ~30% degradation of AMT was observed after 3 h of irradiation in the presence of the scavengers (Figure 4 A). This aligns with the values obtained from the photooxidation of AMT in the presence of MES without the addition of the ROS quenchers (Figure 3A). Therefore, sufficient ROS production requires charge transfer from an excited flavin molecule.

The addition of TEMPO, DABCO and mannitol leads to a 2.5%, 14.09% and 20.17% photodegradation of AMT after 1 h of irradiation in the presence of IONP-DAFL (Figure 4B). This is significantly less than the 91% degradation after 1 h of irradiation without any scavenger (Figure 3A). Therefore, it could be concluded that hydroxyl radicals, superoxide and singlet oxygen, are all formed during the aerobic photodegradation of AMT. However, the photoreactions with both mannitol and DABCO proceed to near completion, while the addition of TEMPO only leads to a 24.76% degradation of AMT in 3 h. Therefore, superoxide radicals are the predominant reactive oxygen species.

### 2.3.2. Amaranth Photoreduction (Anaerobic Activity)

The photocatalytic degradation of AMT was also investigated under anaerobic conditions in an Ar atmosphere. In this case, we hoped to take advantage of the flavin photoredox catalysis via formation of the hydroquinone capable of reducing the azo substrate by hydride transfer [56], as well as a flavin-sensitized charge transfer to IONP and surface interaction of absorbed AMT [61] (see Scheme 4). Similarly, the reaction was conducted in the presence of either MES or EDTA solution as the electron donor.

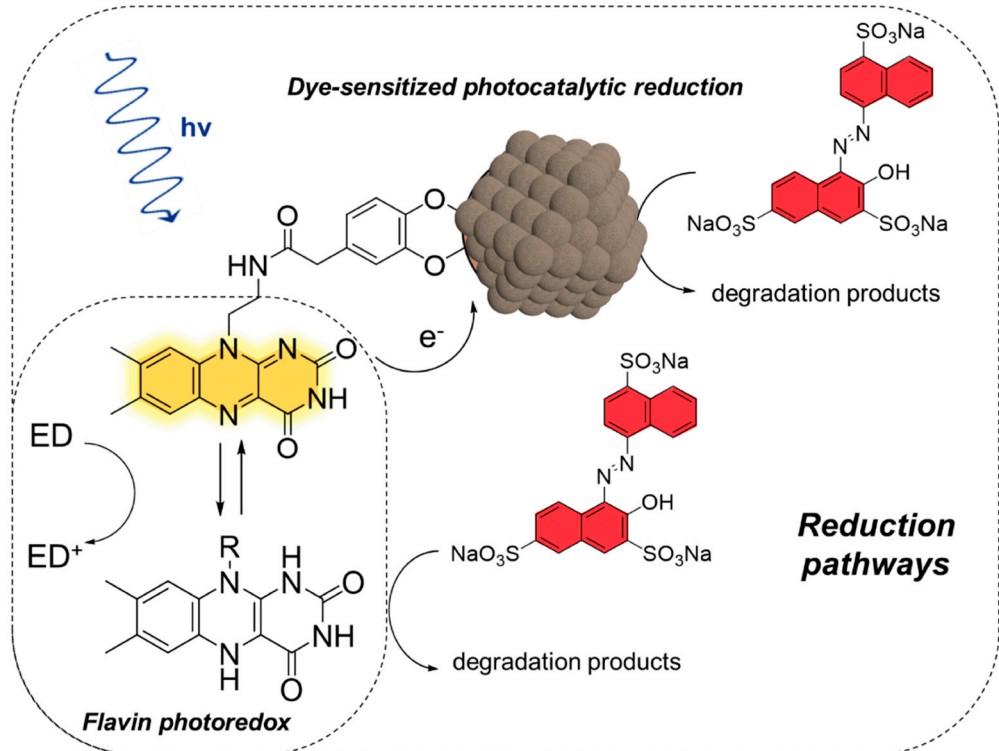

**Scheme 4.** Possible reduction pathways of AMT involving flavin photoreduction of AMT in the presence of an electron donor (ED) and the photo-sensitized reduction of AMT on the IONP surface through charge transfer from the flavin-catechol dye.

After 3 h of irradiation, both bare IONPs and IONP-DA (both 0.333 mg/mL) show negligible activity in MES and EDTA, as seen in previous aerobic experiments. In MES, after 1 h of irradiation, the $C/C_0$ of AMT decreases by 33.4% in the presence of IONP-DAFL (0.333 mg/mL) and by 5.3% in the presence of DAFL (0.003 mg/mL) on its own (Figure 5A).

After 3 h of irradiation, the total AMT degradation is 39.5% and 5.7% in the presence of IONP-DAFL and DAFL, respectively (for UV-vis absorption spectra at each time point refer to Figure S11). However, after 1 h of irradiation in the presence of EDTA (Figure 5B) the $C/C_0$ of AMT decreases by 92.2% in the presence of IONP-DAFL, and by 77.2% in the presence of DAFL (for UV-vis absorption spectra at each time point refer to Figure S12). After 2 h of irradiation, the photodegradation process in the presence of both IONP-DAFL and DAFL is complete. EDTA is therefore a much better electron donor than MES in anaerobic conditions, which could be attributed to the radical EDTA oxidation products that facilitate further AMT degradation [59]. Moreover, it is apparent that, irrespective of the electron donor, IONP-DAFL is a far more efficient photocatalyst than homogenous flavin, clearly demonstrating the benefit of our synergistic photocatalytic system. Refer to the 'Quantum Efficiency' section of the ESI for the rate of the reaction and quantum efficiency calculation.

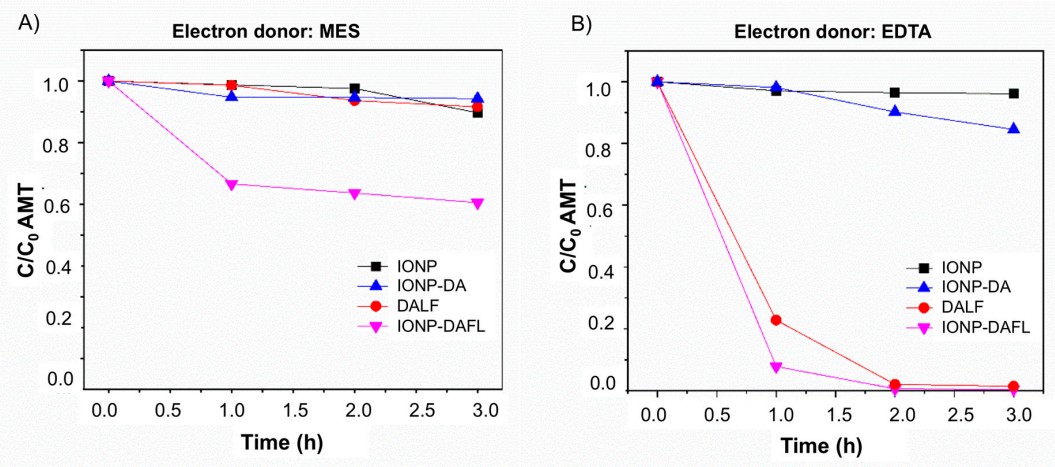

**Figure 5.** The relative concentration of remaining AMT in solution ($C/C_0$) measured using UV-Vis absorption spectroscopy ($\lambda_{max}$ = 520 nm) after irradiation with 450 nm blue light for 3 h in the presence of IONP-DAFL, DAFL, IONP, or IONP-DA in (**A**) MES (0.1 M) and in (**B**) EDTA (0.1 M) under an Ar environment.

Control experiments conducted in the dark in an inert Ar atmosphere with EDTA, and in aerobic conditions with MES as an electron donor (Figure S13) show negligible activity for IONP-DAFL after 8 h in the presence of AMT. Lastly, without the presence of an electron donor, all control samples (IONP, IONP-DA and DAFL) displayed negligible photooxidative activity (Figure S14A). IONP-DAFL, on the other hand, showed 34.02% AMT degradation in $O_2$ (Figure S14A) and a 25.05% degradation in Ar (Figure S14B) after 3 h, which is most likely the result of electron donation from AMT itself, which then leads to its degradation, rather than water acting as the donor.

### 2.3.3. Reusability of the Catalyst

The reusability of the IONP-DAFL in the anaerobic photodegradation of AMT using EDTA (Figure 6A) and the aerobic degradation of AMT with MES (Figure 6B) were investigated. AMT (0.05mg/mL) was irradiated in the presence of IONP-DAFL (0.333 mg/mL) and EDTA or MES (0.1 M) for 1 h under Ar or $O_2$, respectively. The IONP-DAFL was then removed from the solution with a magnet, washed, and re-used in another run. Heterogenous flavin-based photocatalysts have often shown poor recyclability in the literature [62], which is also confirmed in this study. Although the IONP-DAFL system can be easily removed from the solution with the use of a magnet, IONP-DAFL is

not recyclable in either aerobic or anaerobic conditions. For the UV-vis absorption spectra for each run in aerobic and anaerobic conditions, refer to Figure S15A,B, respectively.

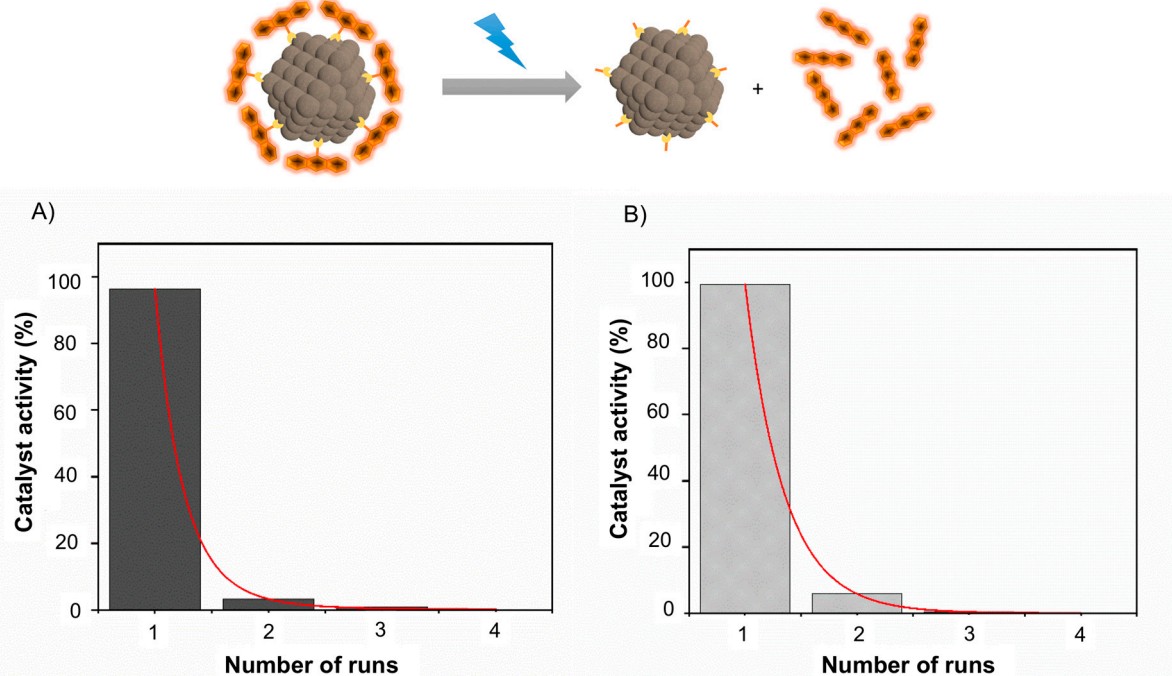

**Figure 6.** The recyclability of IONP-DAFL over four runs of AMT photodegradation with (**A**) MES in $O_2$ and with (**B**) EDTA as an electron donor in Ar. The catalyst's activity is calculated from the concentration of remaining AMT in solution ($C/C_0$) measured using UV-vis absorption spectroscopy ($\lambda_{max}$ = 520 nm) after irradiation with 450 nm blue light for 1 h per run. The images show the flavin molecules separated from the nanoparticle surfaces due to photodegradation.

A significant loss in the catalyst's activity is observed after one run. The low recyclability of the material is most likely due to particle and flavin instability. Flavin photodealkylation at the N10 position has been reported in the literature [33]. In the case of IONP-DAFL, this would release the flavin moiety into the supernatant. To test this hypothesis, two samples of 1.0 mg/mL IONP-DAFL in $H_2O$ were irradiated with a 450 nm blue light in $O_2$ and Ar for 1 h. The particles were then removed, and the fluorescence intensity of the supernatants of the samples were measured (Figure S16). A strong fluorescence signal was obtained from both supernatants after excitation at 450 nm, indicating that flavin species were present in solution and no longer conjugated to the IONP surface after irradiation. Furthermore, a stronger fluorescence signal was obtained from the supernatant of the sample irradiated in $O_2$, thus indicating that IONP-DAFL is less stable when irradiated in an $O_2$-rich environment. Lastly, Figure S6C indicates that the fluorescence signal of DAFL is weak in comparison to NBoc-protected flavin, **3** due to quenching by the catechol moiety. Therefore, the fluorescence signals in Figure S16 can be attributed to lumiflavin that is a product of flavin photodegradation [63]. The regeneration of the catalyst was therefore not possible. For this reason, our current work looks at addressing this issue through investigating different anchoring groups, conjugation strategies and immobilization for flavin IONP hybrids with a wider range of sacrificial electron donors [59] that, unlike MES and EDTA, are stable in nonacid environments and at different temperatures.

## 3. Materials and Methods

### 3.1. General

Commercially available reagents were purchased in the highest purity from Acros Organics (Pittsburgh, PA, USA), Alfa Aeser (Haverhill, MA, USA), Sigma-Aldrich (St. Louis, MO, USA), and TCI Chemicals (Tokyo, Kanto region, JPN) [20]. A Bruker (Billerica, MA, USA) 500 MHz DCH Cryoprobe Spectrometer was used for the $^{13}$C and $^{1}$H Nuclear Magnetic Resonance (NMR) spectroscopy measurements. UV-vis absorption spectra were obtained using an Agilent (Santa Clara, CA, USA) Cary 300 Spectrophotometer. Attenuated Total Reflection Fourier-Transform Infra-Red (ATR-FTIR) spectroscopy data was acquired from powder samples using a Perkin Elmer (Waltham, MA, USA) Spectrum One FT-IR Spectrometer. Fluorescence intensity measurements were done using an Agilent Technologies Cary Eclipse Fluorescence Spectrophotometer. Dynamic Light Scattering (DLS) and Zeta potential measurements were obtained using a Zetasizer Nano Range instrument from Malvern Panalytical (Malvern, Worcs, UK).

Electron microscopy images were taken using a FEI (Hillsboro, OR, USA) 200 kV Tecnai 20 Transmission Electron Microscope (TEM). The samples, dispersed in water, were drop cast and left to air-dry on Agar Scientific Ltd (Stansted, ESX, UK) lacey carbon copper grids [20]. X-ray photoelectron spectroscopy (XPS) measurements were carried out using a Thermo Fisher Scientific (Waltham, MA, USA) Escalab 250Xi ultrahigh-vacuum (UHV) photoemission instrument. X-ray Diffraction (XRD) measurements were conducted using a Bruker D8 Davinci Diffractometer (Cu-K$\alpha$1 source, $\lambda$ = 1.5406 Å).

### 3.2. Synthesis of Ferrous dodecyl Sulfate (Fe(DS)$_2$)

Briefly, a 100 mL solution containing 1 M sodium dodecyl sulfate (SDS) and 1 M iron (II) chloride (FeCl$_2$) in Milli-Q$^{®}$ water is prepared and stored at 2 °C for 1 h. The resulting Fe(DS)$_2$ precipitate is then washed with 2 °C Milli-Q$^{®}$ several times, and allowed to re-crystallize overnight.

### 3.3. Synthesis of $\gamma$-Fe$_2$O$_3$ Maghemite Iron Oxide Nanoparticles (IONPs)

The $\gamma$-Fe$_2$O$_3$ maghemite IONPs were prepared using a modified "one emulsion plus reactant" [64] micro-emulsion synthesis protocol described by Benyettou et al. [36]. 3.05 g of Fe(DS)$_2$ is dissolved in a 500 mL round-bottom flask containing 345 mL of Milli-Q$^{®}$ water at 32 °C. Once the iron salt is dissolved, the temperature is reduced to 28 °C. 50 mL (40 wt%) dimethylamine, heated to the same temperature, is then added dropwise. The solution is left shaking for 2 h at the same temperature without the use of a magnetic stirrer. The flask is then placed on ice. The nanoparticles are then cleaned 10 times with Milli-Q$^{®}$ water using a magnet. 4 mL of 4 M HCl is added before each wash for the first fourwashes. The IONPs are then stored in water at a pH of 7.

### 3.4. Synthesis of Dopamine-Flavin (DAFL)

Details on the synthesis of DAFL can be found in the Supplementary Information section.

### 3.5. Surface Modification of $\gamma$-Fe$_2$O$_3$ Nanoparticles

Briefly, 5 mg of IONPs are suspended in a 5 mL solution of dopamine (DA) or dopamine-flavin (DAFL) (0.5 mg/mL in H$_2$O) in an argon atmosphere at 23 °C. The solution is sonicated for 5 min. A vortex mixer is then used to shake the suspension for 2 h, while being protected from direct sunlight. The suspension is then washed 3–4 times on a magnet using Milli-Q$^{®}$ water. The coated IONPs are kept in the dark at room temperature in air.

*3.6. General Procedure for Photoreduction and Photooxidation of Amaranth*

AMT (0.05 mg/mL) and an IONP sample (0.333 mg/mL) or DAFL (0.003 mg/mL) were added to a solution of either MES or EDTA buffer (0.1 M, pH 6, 3 mL), and then saturated with either $O_2$ gas or Ar for 15 min before being irradiated with a 450 nm blue LED (18 W; 34 mW/cm$^2$) in a HepatoChem (Beverly, MA, USA) EvoluChem™ PhotoRedOx Box photoreactor, equipped with a cooling fan to keep the temperature at ~23 °C. 100 μL aliquots were removed from the reaction at specific time points and diluted to 1 mL with 0.1 M MES/EDTA. Any nanoparticles were removed by centrifugation or using a magnet before monitoring the degradation of AMT by UV-vis absorption spectroscopy, using $\lambda_{\text{max}}$ = 520 nm of AMT [8] to observe the relative concentration of remaining AMT in solution ($C/C_0$), where $C_0$ represents the initial concentration of AMT before irradiation.

*3.7. General Procedure for ROS Scavenging Experiments*

Mannitol, TEMPO or DABCO (1.5 equiv) were added to a solution of AMT (0.1 mg/mL), IONP-DAFL or IONP (0.333 mg/mL) in MES buffer (0.1 M, pH 6, 3 mL). The mixture was then saturated with $O_2$ gas and irradiated with a 450 nm blue LED (18 W; 34 mW/cm$^2$) in an EvoluChem™ PhotoRedOx Box photoreactor equipped with a cooling fan to keep the temperature at ~23 °C.

*3.8. General Procedure for Recyclability Measurements*

The recyclability of IONP-DAFL was investigated using the general procedure described in Section 3.6 using MES buffer (0.1 M, pH 6) in an aerobic environment and with EDTA buffer (0.1 M, pH 6) under anaerobic conditions. One time point is taken at 1 h for each run before the catalyst was removed from the reaction via magnetic separation and washed with Milli-Q® $H_2O$ for reuse. Four cycles for both samples were completed.

## 4. Conclusions

We have developed a new hybrid metal oxide-organic heterogeneous photocatalytic system capable of degrading azo bonds found in industrial dyes such as AMT. The hybrid flavin-IONP system (IONP-DAFL) is shown to enable both the photooxidation and photoreduction in an $O_2$ and Ar medium, respectively. While controls were found to be not photocatalytically active, IONP-DAFL and flavin-dopamine (DAFL) alone were shown to reduce AMT, with best results obtained in the presence of EDTA and in inert atmosphere. This was attributed to the synergistic action of the flavin-mediated photocatalysis and IONP photoexcitation. Lack of activity in IONPs controls in Ar can be explained by fast electron-hole recombination that commonly occurs in photoexcited IONPs [29].

In an $O_2$ environment, IONP-DAFL and DAFL were able to degrade the AMT dye in both electron donor buffers (MES and EDTA) through the release of ROS species shown to predominantly be superoxide radicals. The activity of IONP-DAFL was higher in MES, most likely due to spin ion pair correlation which prevents the flavin from degradation [55]. Contrary to IONP-DAFL, the photocatalysis of DAFL alone was more efficient in EDTA, probably due to an additive effect of EDTA's reactive oxidation products. The control photocatalysts, IONP and IONP-DA, displayed little catalytic activity both in EDTA and MES.

In summary, we designed a novel hybrid photocatalyst that combines iron oxide nanoparticles and flavin moieties anchored to the nanoparticle surface using catechol linker. The catalysts mimic the activity of naturally-occurring azoreductatese enzymes, and facilitates azo-bond reduction in inert atmosphere. Interestingly, this activity can be switched to oxidation by the introduction of oxygen, and in air the hybrid mimics the activity of oxidases, such as laccase, which can oxidize the azo bond [65]. Moreover, unlike its enzymatic counterparts, the hybrid system is robust, and can be stored at room temperature over longer time, and can be efficiently removed from solution using a magnet, although not reused. Limited recyclability is attributed to the low stability of the flavin-IONP linkage, and our

current efforts are focused on the redesign of the hybrid system by use of improved anchoring linkers and immobilization of additional electron donors.

However, despite limited recyclability, photoactivation with blue light enables temporal and spatial control over the reaction, and the hybrid system is nontoxic, cheap, easy to prepare, and easy to remove from the solution. Therefore, we believe that it has a significant potential to be used in water remediation, and our future work with a look into the degradation of other contaminants, such as rhodamine B (RhB) and bisphenol A (BPA) [66], and the design of a suitable flow cell/membrane system, suitable for practical applications.

**Supplementary Materials:** The following are available online at http://www.mdpi.com/2073-4344/10/3/324/s1, Figure S1: X-ray diffraction spectroscopy pattern of the -Fe2O3 IONPs; Figure S2: The ATR-FTIR spectrum of -Fe2O3 IONPs functionalized with dopamine (IONP-DA) in comparison to IONP and dopamine (DA); Figure S3: TEM images of IONP-DA at a scale of (A) 100nm and (B) 20nm. (C) The particle size distribution determined from the average of ~100 measurements of particles; Figure S4: (A) XPS survey of IONP-DAFL and associated spectra: (B) Fe 2p high resolution scan, (C) O 1s high resolution scan, (D) N 1s high resolution scan, and (E) C 1s high resolution scan; Figure S5: X-ray diffraction spectroscopy pattern of the IONP-DAFL in comparison to IONP; Figure S6. (A) Molecular structure of NBoc-flavin (3) and DAFL, (B) their UV absorption spectrum, and (C) fluorescence emission intensity; Figure S7: UV-Vis absorption spectroscopy measurements of AMT ($\lambda$max= 520nm) after irradiation with 450 nm blue light for 3 hours in the presence of (A) IONP, (B) IONP-DA, (C) DAFL, or (D) IONP-DAFL in MES (0.1 M) under an O2 atmosphere; Figure S8: UV-Vis absorption spectroscopy measurements of AMT ($\lambda$max= 520nm) after irradiation with 450 nm blue light for 3 hours in the presence of (A) IONP, (B) IONP-DA, (C) DAFL, or (D) IONP-DAFL in EDTA (0.1 M) under an O2 atmosphere; Figure S9: UV-Vis absorption spectroscopy measurements of AMT ($\lambda$max = 520 nm) after irradiation with 450 nm blue light for 3 hours in the presence of (A) IONP+TEMPO, (B) IONP+DABCO, or (C) IONP+mannitol in MES (0.1 M) under an O2 atmosphere; Figure S10: UV-Vis absorption spectroscopy measurements of AMT ($\lambda$max = 520 nm) after irradiation with 450 nm blue light for 3 hours in the presence of (A) IONP-DAFL+TEMPO, (B) IONP-DAFL+DABCO, or (C) IONP-DAFL+mannitol in MES (0.1 M) under an O2 atmosphere; Figure S11: UV-Vis absorption spectroscopy measurements of AMT ($\lambda$max = 520 nm) after irradiation with 450 nm blue light for 3 hours in the presence of (A) IONP, (B) IONP-DA, (C) DAFL, or (D) IONP-DAFL in MES (0.1 M) under an Ar atmosphere; Figure S12: UV-Vis absorption spectroscopy measurements of AMT ($\lambda$max = 520 nm) after irradiation with 450 nm blue light for 3 hours in the presence of (A) IONP, (B) IONP-DA, (C) DAFL, or (D) IONP-DAFL in EDTA (0.1 M) under an Ar atmosphere; Figure S13: UV-Vis absorption spectroscopy measurements of AMT ($\lambda$max = 520 nm) after being kept in the dark for 8 hours in the presence of (A) IONP-DAFL in MES (0.1 M) under an O2 atmosphere and (B) IONP-DAFL in EDTA (0.1 M) under an Ar atmosphere; Figure S14: (A) The relative concentration of remaining AMT in solution (C/C0) measured using UV-vis absorption spectroscopy ($\lambda$max = 520 nm) after irradiation with 450nm blue light for 3 hours in the presence of IONP, IONP-DA, DAFL, or IONP-DAFL in the absence of an electron donor (ED) in an O2 atmosphere. (B) The relative concentration of remaining AMT in solution (C/C0) measured using UV-vis absorption spectroscopy ($\lambda$max = 520 nm) after irradiation with 450 nm blue light for 3 hours in the presence of IONP-DAFL in the absence of an electron donor (ED) in comparison to EDTA as an electron donor (ED) under an inert Ar environment; Figure S15: UV-Vis absorption spectroscopy measurements of AMT ($\lambda$max = 520 nm) after irradiation with 450 nm blue light for 1 h per run for 4 runs in the presence of (A) IONP-DAFL in MES (0.1 M) under an O2 atmosphere and (B) IONP-DAFL in EDTA (0.1 M) under an Ar atmosphere; Figure S16: Fluorescence intensity measurements of aliquots taken from the supernatant of IONP-DAFL after irradiation with blue light in O2 or Ar for 1 h ($\lambda$ex = 450 nm),Table S1: DLS and zeta potential measurements of IONP, IONP-DA, and IONP-DAFL samples. The errors are calculated from the standard deviation of 3 repeats. Sizes from TEM images are the average of the particle size distribution of 100 particles of each sample. The error is the standard deviation.

**Author Contributions:** Conceptualization, L.C., S.I.N. and L.F.; investigation, S.I.N. and L.C.; data curation, S.I.N. and L.C.; writing—original draft preparation, S.I.N.; writing—review and editing, L.C. and L.F.; funding acquisition, L.F. All authors have read and agreed to the published version of the manuscript.

**Funding:** This research was funded by EPSRC DTC Studentship, grant number EP/N509620/1.

**Acknowldgments:** We wish to thank the staff of NMR and MS facilities in the Department of Chemistry, the University of Cambridge for their assistance, the EM suite staff in the Department of Physics, University of Cambridge for their help with TEM imaging, and Adam Brown of the Maxwell Centre, University of Cambridge for his assistance with XPS measurements.

**Conflicts of Interest:** The authors declare no conflict of interest.

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
