# Peer review of "Flavin-Conjugated Iron Oxide Nanoparticles as Enzyme-Inspired Photocatalysts for Azo Dye Degradation"

_catalysts, doi:10.3390/catal10030324_

Round 1

Reviewer 1 Report

This article summarized author's work in using magnetic nanoparticle immobilized enzymes for photodegradation of dyes. While many photocatalysts related articles focus on using inorganic semiconductor materials, this article brings novelty in using enzyme based material for actual reaction while using nanoparticle as a support material for ease of recycling. The ideas are novel and the experiments are well designed and executed. I think this article is interesting to other people in photocatalyst field.

Author Response

We thank the reviewer for the positive comments concerning the article.

Reviewer 2 Report

The manuscript reports constructing a conjugate system of iron oxide and an organic ligand for water purification purposes. This might be of interest. However, generally organic ligands get finally oxidized during photoreactions especially under the sunlight. A point to address is also to provide an estimate of quantum efficiency for the degradation of the dye. To do this, light flux entering the photoreactor should be measured and included. 

There are some minor errors in the text like incorrect or missing dimensions and numbers. So the manuscript should be read very attentively and corrected. 

Author Response

We thank the reviewer for the comments and we agree that this is not a perfect system, and are currently working on improved linker system that stabilises flavin significantly. This paper provides guiding ideas to develop such recyclable catalysts in the future, and could be useful for some applications within different carrier systems ( not in solution where degradation is a problem). 

We also agree that more information should be given about the quantum efficiency and this has now been included within the ESI. 

Reviewer 3 Report

The article "Flavin-conjugated Iron Oxide nanoparticles as Enzyme-inspired Photocatalyts for Azo Dye Degradation"submitted to Catalysts is well written and concerns very current topics in  catalysis and materials chemistry. It is written clearly and comprehensibly without unnecessary information. The obtained results are clear and the conclusions  are correct. However, before the article will be published, some details needs clarification.

1. A schematic figure with nanoparticles with marked layers on the surface of maghemite could be useful (reading the work, you really forget what's  the shell covering the magnetic core).

2. It is unclear whether the authors present FTIR or ATR-FTIR spectra. This must be clearly indicated in the Methods section. If this is the FTIR spectrum, Authors must describe how the KBR pellet was made or what dispersion medium (nujol, HCB ..) was used. 

3. It is better to show IR spectra in Absorbance rather than Transmittance, which is already a calculated value, but this is really a small note.

4. In line 255 one should mention what radiation was used (wavelength and intensity). Both values are given below the charts and in the experimental section, but providing them here will save the reader the search.

5. The results of the reuse of the catalytic system are the most questionable. After the first cycle, the catalyst is almost inactive so it is rather difficult to talk about its reuse. Did the authors try to regenerate the catalyst somehow after the cycle, because it does not result directly from the text. In addition, in my opinion, the effect of pH and temperature on the catalytic activity of the system should be examined in order to fully characterize it.

Author Response

We thank the reviewer for positive comments concerning the content and the presentation of the paper, and really appreciate the additional comments on the aspects that could be improved. 

We have addressed them as follows:

1. A schematic figure with nanoparticles with marked layers on the surface of maghemite could be useful (reading the work, you really forget what's  the shell covering the magnetic core)

 We have simplified the scheme 2 to clearly show the layer of flavin-dopamine on the surface. 

2.It is unclear whether the authors present FTIR or ATR-FTIR spectra. This must be clearly indicated in the Methods section. If this is the FTIR spectrum, Authors must describe how the KBR pellet was made or what dispersion medium (nujol, HCB ..) was used. 

ATR-FTIR was used for the experiments and we have included this information in the text. 

3. It is better to show IR spectra in Absorbance rather than Transmittance, which is already a calculated value, but this is really a small note.

  We have changed the spectra accordingly.

4. In line 255 one should mention what radiation was used (wavelength and intensity). Both values are given below the charts and in the experimental section, but providing them here will save the reader the search.

We have added this information as suggested.

5.The results of the reuse of the catalytic system are the most questionable. After the first cycle, the catalyst is almost inactive so it is rather difficult to talk about its reuse. Did the authors try to regenerate the catalyst somehow after the cycle, because it does not result directly from the text. In addition, in my opinion, the effect of pH and temperature on the catalytic activity of the system should be examined in order to fully characterize it.

We agree that this is the weakest point of the manuscript, simply as the catalyst containing this particular modification cannot be recycled and reused over more than one run. This was maybe not stated clearly enough in the original manuscript, and the figure which was used to illustrate this was not clear either. The text and the figure were now imodified to emphasize that the system is non-recyclable.

In terms of catalyst regeneration, we state in lines 405-413 that fluorescence studies on IONP-DAFL after irradiation show that the flavin is released from the nanoparticle surfaces and is converted to lumiflavin. Catalyst regeneration was thus not possible. We have now edited the text to further explain what is likely occurring. Flavin photodealkylation at the N10 position has been reported in the literature. Dopamine modification of flavin in our system is achieved through  N10 position. It is therefore likely that  dealkylation causes detachment of  the flavin thus leaving behind the iron oxide nanoparticles with dopamine on the surface. One of the approaches we have taken to address this problem is designing flavins with anchoring groups at the N3 position instead. This has shown to improve stability. Moreover, as mentioned, different anchoring groups, such as phosphates instead of dopamine, that have a higher affinity to the IONPs will be used.

Lastly, the effects of pH and temperature were not assessed in this reaction since the buffers used – MES and EDTA – have been shown to be only stable at acidic pH and short temperature ranges. These buffers were chosen because the literature has shown them to be some of the most effective electron donors and stabilizers for Flavin moieties. In our future work, we plan on using different buffers that operate at different pH and temperature ranges